# Organic Molecules: Is It Possible to Distinguish Aromatics from Aliphatics Collected by Space Missions in High-Speed Impacts?

**Mark Burchell ***  **and Kathryn Harriss**

Centre for Astrophysics and Planetary Science, School of Physical Sciences, University of Kent, Canterbury, Kent CT2 7NH, UK; K.Harriss@kent.ac.uk

* Correspondence: m.j.burchell@kent.ac.uk; Tel.: +44-1227-827833

**Abstract:** A prime site of astrobiological interest within the Solar System is the interior ocean of Enceladus. This ocean has already been shown to contain organic molecules, and is thought to have the conditions necessary for more complex organic biomolecules to emerge and potentially even life itself. This sub-surface ocean has been accessed by Cassini, an unmanned spacecraft that interacted with the water plumes ejected naturally from Enceladus. The encounter speed with these plumes and their contents, was between 5 and 15 km s$^{-1}$. Encounters at such speeds allow analysis of vapourised material from submicron-sized particles within the plume, but sampling micron-sized particles remains an open question. The latter particles can impact metal targets exposed on the exterior of future spacecraft, producing impact craters lined with impactor residue, which can then be analysed. Although there is considerable literature on how *mineral* grains behave in such high-speed impacts, and also on the relationship between the crater residue and the original grain composition, far less is known regarding the behaviour of *organic* particles. Here we consider a deceptively simple yet fundamental scientific question: for impacts at speeds of around 5–6 km s$^{-1}$ would the impactor residue alone be sufficient to enable us to recognise the signature conferred by organic particles? Furthermore, would it be possible to identify the organic molecules involved, or at least distinguish between aromatic and aliphatic chemical structures? For polystyrene (aromatic-rich) and polymethylmethacrylate (solely aliphatic) latex particles impinging at around 5 km s$^{-1}$ onto metal targets, we find that sufficient residue is retained at the impact site to permit identification of a carbon-rich projectile, but not of the particular molecules involved, nor is it currently possible to discriminate between aromatic-rich and solely aliphatic particles. This suggests that an alternative analytical method to simple impacts on metal targets is required to enable successful collection of organic samples in a fly-by Enceladus mission, or, alternatively, a lower encounter speed is required.

**Keywords:** astrobiology; Enceladus; space-missions; organic; aromatic; aliphatic

## 1. Introduction

The search for evidence of life in the Solar System beyond the Earth used to have the mantra—find the water. The idea was that water is a key ingredient of life as we understand it, so a suitable habitat for life would be one that had liquid water readily available. A key step in the search for life would be to identify potential habitats and to focus the search there. Thus, the presence of liquid water would identify sites worth exploring. However, we now know of many Solar System bodies, which possess liquid water in such quantities, these are classed as ocean worlds [1]. These bodies are typically icy

satellites of outer planets, where the liquid water is not on the surface, but in the interior under a surface cap of ice. Liquid water alone is not sufficient. There also needs to be chemical disequilibria (or to put it another way a source of energy to drive chemical reactions) and the presence of organic molecules to provide the inputs to drive forward the prebiotic chemistry and then the increase in complexity characteristic of life. As discussed further below, the icy ocean worlds may meet these criteria so are considered of great interest in terms of the search for life elsewhere.

How to access these sites to explore their contents is thus an issue. Remote sensing of the surface ice can provide some clues as to the composition of the interior oceans. However, conveniently, some of these icy satellites naturally expel water into space via plumes. Europa and Enceladus are particularly good examples, and the plumes permit a more direct sampling of the contents of the oceans. A proper characterization of these plumes focusses on determining their content in detail. For example, what is the mineral content? In addition, is there any evidence for organics?

Organic materials are widespread in space, for example, a notable fraction of the carbon in the interstellar medium is in the form of polycyclic aromatic hydrocarbons, e.g., [2,3]. Aliphatic organic compounds have also been observed, not just in the interstellar medium but also on asteroids [4] and comets for example [5]. We thus should expect to find organics when we visit places in the Solar System where chemistry has processed materials.

One simple question is: Can we recognise the organic material in an in situ analysis? Further, even if we cannot identify the exact nature of the material, can we do something as basic as distinguish aromatics from aliphatic? As ever with space missions, the answer depends not just on the analysis method, but also on the sample collection method, which may cause damage to the samples. This paper briefly considers one aspect of this by looking at Enceladus, an icy satellite of Saturn.

*Enceladus and Sampling Its Plumes*

There is now ample evidence that Enceladus contains an interior ocean, e.g., [6–11]. Further, due to gravitational forces arising from its orbit around Saturn, water plumes are forced out of the icy surface of Enceladus near its southern polar regions. These plumes were first observed by the Cassini spacecraft orbiting Saturn e.g., [12–14]. By examining the material from the plume, the Enceladan internal ocean has been shown to be salt rich [15]. If we accept that the centre of Enceladus is a solid core with a rocky, mineral content, the ocean floor will be in contact with minerals. Due to heat added by the gravitational flexing processes during the orbit around Saturn, plus any internal heat in the core, the ocean will be warm. Water, heat, minerals and salts are all key ingredients for interesting chemistry, and maybe something more, biology perhaps? Conveniently, a visiting space mission can sample this fascinating ocean by flying through the ejected plume. This minimises the risk of contamination of Enceladus, thus preserving its isolated nature (unless the spacecraft unfortunately crashes into the ice) and so observing the necessary planetary protection protocols.

A visiting spacecraft has the option of orbiting the parent planet Saturn (easier, plus more science can be done elsewhere in the Saturnian system), or of entering orbit around the satellite itself. In both cases, it will have to pass close to the surface, at an altitude of much less than 100 km, in order to intercept the plume before the larger droplets of water (which may have frozen into ice) fall back to the surface. Indeed, the lower the altitude the better in terms of droplet size. Which approach is taken is important as it dictates the relative speed of the craft when it intercepts the plume. If it is in a Saturnian orbit the encounter speed will be many km s$^{-1}$, if it orbits the satellite it could be as low as several hundred m s$^{-1}$. Data has been obtained in situ from the Enceladus plumes by the Cassini space mission, which flew through the water vapour plumes as it passed by Enceladus. Results on the Enceladus plume composition are given in [16,17], where the various Saturnian orbits followed by Cassini allowed Enceladus fly-by data to be collected at impact speeds of 5–15 km s$^{-1}$, with typical impacts being in the speed range 6–8 km s$^{-1}$. The data in [16,17] show the presence of organic molecules in the brine.

To investigate further we could mount a sample return mission to Enceladus. This is discussed for example in [18], where it is noted that planetary protection would impose enormous cost and

complexity into such a mission. Indeed, as noted in [19], planetary protection regarding Enceladus would require that there is no direct physical link between the source material and the general environment here on Earth. One way to avoid this is to sample and analyse in situ.

Here, let us imagine a mission, which flies past Enceladus, collects samples and analyses them in situ. We will consider the worst case of the higher speed impact as being that which occurs, i.e., sample collection occurs when a spacecraft passes Enceladus whilst orbiting Saturn. The data in [16,17] from Cassini were obtained by impact ionization during such fly-bys. In such a method, small (sub-micrometre) grains are vapourised during the impact, and the ionic plasma that formed was measured in a time of flight system. As noted in the supplemental material to [16], and previously in papers such as [20–23], at the encounter speeds during the Cassini fly-bys of Enceladus ($<20$ km s$^{-1}$), the impacting materials are not reduced to their elemental composition. Instead, molecular fragments are formed. These fragments have mass numbers (assuming single ionization) which show regular spacings in mass whose nature differs depending on the chemistry of the sample, and, for example, if the sample was originally an aromatic or aliphatic compound. We can thus say that it is possible to differentiate between different types of organic compounds. Nevertheless, this does not positively identify the sample, it just gives mass numbers of the fragments formed in the impact process. However, an ideal analysis would go further. We would want to determine what compound was it originally, and know more about its structure. The normal way to do this is to collect macroscopic residue from the impacting particle, and to analyse that.

For a macroscopic dust sample (and here micrometre scale is macroscopic), there is a wealth of data concerning how mineral grains behave in impacts at speeds up to 6 km s$^{-1}$, e.g., [24–28]. A metal plate or foil, can act as a target. An impact crater is formed when the dust grain strikes the target, and impact residue lines the crater. The shock pressures in the sample and target can be in the tens of GPa, e.g., [29]. There will be some heating as the sample releases from its shocked state and indeed some melt may form, but some samples may retain their original crystalline structure (e.g., see [30–32]). Analysis of the resulting residue then reveals information about the impactor. A suitable analysis method has to be found, the common ones used in the laboratory include scanning electron microscopy with elemental analysis via dispersive X-rays (EDX-SEM). Structural information in the laboratory can come from either Raman spectroscopy or TEM work on samples. All this equipment tends to be bulky so is not ideally suited for a space mission where size, mass and power are major constraints. We can however suppose that a suitable technique will be found. For example, Raman spectrometers have been made robust enough to be deployed on space missions and two will soon be sent to the surface of Mars (on NASA's Mars 2020 mission and ESA's future ExoMars rover mission).

So can we now imagine that we can analyse the samples fully? Unfortunately, some issues remain. If thermally robust mineral grains are all that is in the water, the analysis is relatively straight-forward, in that the impact process may break particles apart, but will often leave their basic nature intact. Unfortunately, some of the more hydrated minerals are the ones that suffer the most in such impacts, so we need to allow for this in the analysis. Worse, organic materials will suffer thermally, and do so depending on their nature. It has previously been shown, in laboratory experiments, that various organic molecules frozen in ice can survive impacts on a variety of targets including water, sand and ice at impact shock pressure up to 10 GPa [33]. However, the results in [33] showed that not all organic compounds survive in equal quantities.

For a future space mission we would likely use metal targets as collecting surfaces, so this needs to be investigated. Accordingly, we present here results for impacts of small organic grains on metal targets, at speeds close to 5 km s$^{-1}$ (the minimum speed in the Cassini fly past of Enceladus). We look at both polystyrene (aromatic-rich) and polymethlymethacrylate (solely aliphatic) projectiles, to see if we can find residues and distinguish between an aromatic and an aliphatic organic impactor respectively.

## 2. Materials and Methods

The experiments were carried out using the two-stage light gas gun of the University of Kent (UK) [34]. The gun was used to fire small particles at speeds around 5 km s$^{-1}$ onto aluminium targets (grade Al-1080). Aluminium was chosen as the target as it has been widely used in past experiments studying impacts of small grains onto metals, and was a collector material in the NASA Stardust mission to collect cometary dust [29]. In each shot a small quantity of projectiles were loaded into a nylon sabot, which was accelerated in the barrel of the gun and then discarded in-flight. A cloud of the projectiles then hits the target, producing impact craters lined with residue. The speed was measured in flight by passage of the particles past laser light stations (see [34] for details).

Two shots were done as part of this work. The projectiles were small spheres of polystyrene (PS, $(C_8H_8)_n$) or polymethlymethacrylate (PMMA, $C_5H_9O_2$), see Figure 1. The projectiles were of mean size 20 ± 1 μm in both cases. This is a slightly larger size compared to what is expected in the plumes of Enceladus. For example, the particles observed in the Cassini fly-bys and reported in [16,17] were typically less than 6 or 7 μm in diameter, with many less than 1 μm size, and in [35] it is suggested that grains of 4–6 μm in size may be present in the plume at 50 km altitude above the surface.

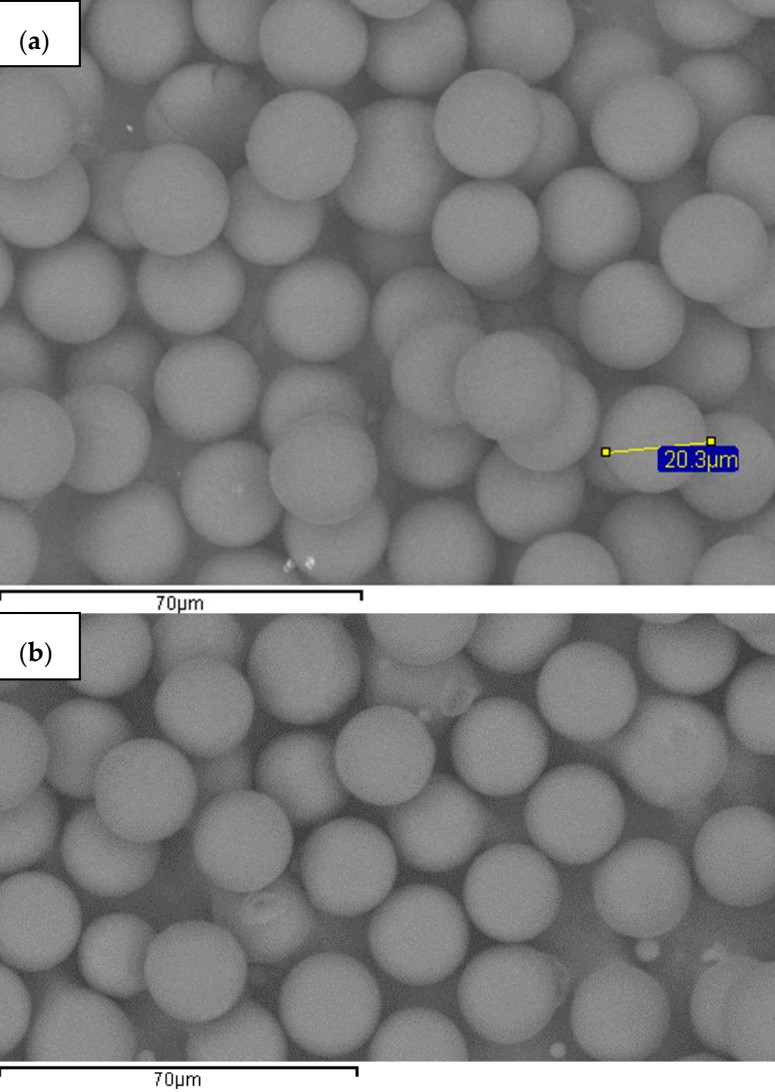

**Figure 1.** Back scattered electron images of the raw projectiles before a shot (**a**) PS and (**b**) PMMA.

The impact speeds were 4.6 km s$^{-1}$ (PS) and 4.85 km s$^{-1}$ (PMMA). The targets were then imaged in a scanning electron microscope (Hitachi S3400N), with electron dispersive X-ray (SEM-EDX) facility (Oxford Instruments X-Max 80 mm$^2$, analysed with INCA software) to identify elements. Impact craters were seen in both cases (see Figure 2 for examples). The EDX spectra for the raw grains both showed not only the expected carbon peaks, but also a strong oxygen peak for PMMA and a weak oxygen peak for PS. An oxygen peak in the PS sample was a surprise, as it is not present in pure PS, but its presence even as a contaminant means it is not possible to distinguish the two materials here based on their EDX spectra. The samples were then imaged in a Horiba LabRAM HR Evolution Raman spectrometer (600 lines per mm grating, green laser 532 nm). Point Raman spectra were taken from residue-rich parts of craters. The beam spot size was about 1–2 μm. We examined 7 craters for PS and 10 for PMMA, seeing similar results each time.

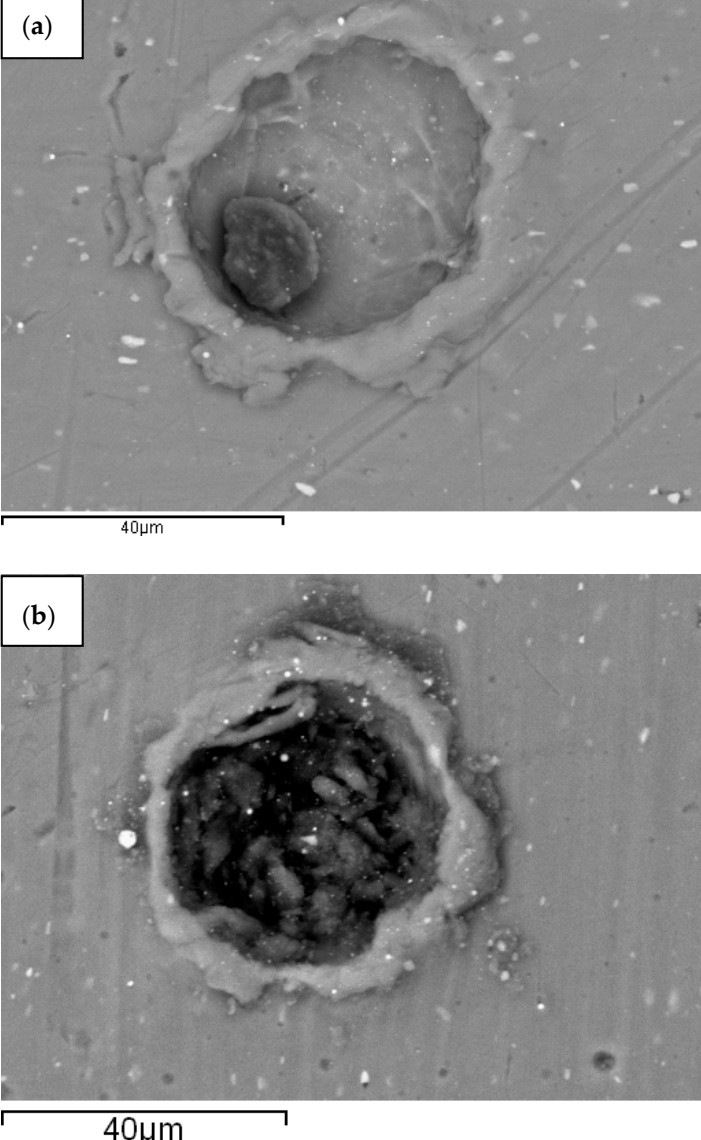

**Figure 2.** Examples of back scattered electron images of impact craters for (**a**) PS at 4.60 km s$^{-1}$ and (**b**) PMMA at 4.85 km s$^{-1}$.

## 3. Results

The Raman spectra (Figure 3) from the raw grains show the standard spectra expected for the materials involved. The raw PS spectrum (Figure 3a) show the expected features for PS [36], with the

strongest peak at 1001 cm$^{-1}$ (the aromatic "breathing mode"), a peak at 1604 cm$^{-1}$ (C=C bonding) and the triplet seen around 3000 cm$^{-1}$ (2852, 2904 and 3054 cm$^{-1}$) associated with C-H stretching. The detail in the spectrum in the range below 1700 cm$^{-1}$, agrees well with those reported previously for PS microparticles [37]. The spectrum of polystyrene is indeed so distinct and so well studied, it is sometimes used to calibrate Raman systems (e.g., ASTM E1840). For the PMMA, the raw grains (Figure 3b) show the expected peaks [38], with the strongest being the feature at approx. 2957 cm$^{-1}$, associated with C-H stretching. The spectra for PS and PMMA are quite distinct.

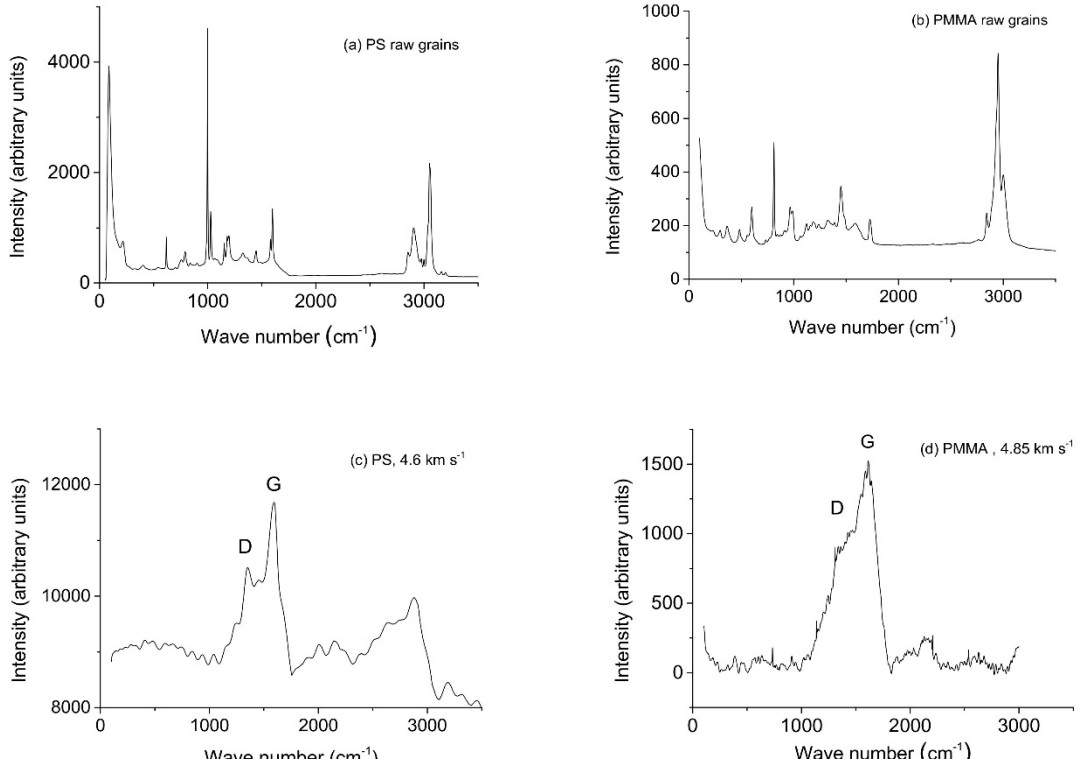

**Figure 3.** Raman spectra of PS and PMMA raw projectiles and crater residues. (**a**) PS raw (average of 5 grains), (**b**) PMMA raw (average of 10 grains), (**c**) PS residue at 4.6 km s$^{-1}$ (average of 7 craters). (**d**) PMMA residue at 4.85 km s$^{-1}$ (average of 3 craters).

In Figure 3, we also show example spectra from residue lining the impact craters for both PS (Figure 3c) and PMMA (Figure 3d). For both PS and PMMA, Raman spectra have been heavily altered from that of the raw materials, with the Raman D and G bands for carbon dominating. These two bands, respectively referring to "disorder" and "graphite" bands, are characteristic of disordered carbonaceous material such as kerogen, soot or burned paper (e.g., [39]). This Raman signal is typical of carbonized organic matter. There does appear to be a broad band just below 3000 cm$^{-1}$ wavenumber for PS, but this by itself cannot be associated with an aromatic rather than an aliphatic compound, nor is it associated with any particular band in the original spectrum. Unlike the D and G bands, none of the other broad bands appears in all the spectra obtained, and may be artefacts of baseline smoothing. It is thus not possible to associate the spectra with specific materials, or to separate out aromatic from aliphatic.

## 4. Discussion

The problem arising from capturing small grains at high speed is that energy is deposited into the samples during the impact process. During the impact, particles are shocked to a high pressure (GPa scale) and temperature. We estimate the peak pressure using the Planar Impact Approximation [40]:

$$P = \varrho Uu, \tag{1}$$

where $\varrho$ is density, U is shock speed and u is particle speed. This uses a linear wave speed equation for each of the materials involved, of the form:

$$U = C + Su, \tag{2}$$

where C and S are material dependent constants. For aluminium we use C = 2707 m s$^{-1}$ and S = 1.356, and for PMMA we use C = 2766 m s$^{-1}$, and S = 1.365 [41]. This gives peak shock pressures of 24.4 GPa (PS) and 18.4 GPa (PMMA).

The temperature rise depends on the energy density in the sample; it is the energy deposited over the whole mass of the grain that is important. Thus the temperature increase is not directly related to the kinetic energy, but to the peak shock pressure of the grain. Consider the case where an incident particle of mass $m$ has speed $v$. The kinetic energy is thus $\frac{1}{2}mv^2$. Simple models suggest half the incident energy is retained in the projectile in the impact (see [42]). So the available energy to heat the particle is $\frac{1}{4}mv^2$. The specific heat capacity of the material can be written as $C_p$, in J kg$^{-1}$ K$^{-1}$. Hence the temperature increase in K is given by energy/(mass × specific heat capacity), which is here $\frac{1}{4}v^2/C_p$. Taking $C_p$ as 1110 J kg$^{-1}$ K$^{-1}$ for PS and 1270 J kg$^{-1}$ K$^{-1}$, for PMMA, and an impact speed of 5 km s$^{-1}$ as here, gives temperature increases of 5600 K for PS and 4900 K for PMMA. These increases are well above the melting points of PS and PMMA (513 and 433 K respectively). This calculation of the temperature increase is a simple one but has been used before as a guide to possible impact temperatures [43]. Nevertheless, it does have problems. For example, it is sensitive to the assumption of how the impact energy is partitioned between the projectile and the target, and ignores latent heat of melting or vaporisation. But it does serve to show that the potential temperatures achieved are an order of magnitude greater than required to melt the particles.

That the impact craters observed here are thus lined with melt should not be a surprise. Unless an organic material has a very high melting point, impacts of grains at 5 km s$^{-1}$, will not yield un-melted samples. This is in contrast with mineral grains in similar impacts, where un-melted impact residue has been found in craters [30–32]. Again, simple models would have predicted that the elevated temperatures due to impact would have melted the whole samples, yet this does not occur for many minerals. This suggests that the simple models either over-estimate the peak post-shock temperature (for example by neglecting latent heat), or the whole impacting grain is not uniformly shocked, leaving macroscopic regions un-melted. In the present case, whilst the temperature increase may be over-estimated, or the whole samples may not be uniformly shocked, the melting points of the organic materials may still be too low to permit un-melted material to survive.

The results do however, show that the impacts of small organic grains produce impact craters on metal targets, which contained significant amounts of impactor residue. Unfortunately, in both cases the impact event has processed the materials to such an extent that they no longer have structures representative of their original state. The results show that we can flag the presence of a high carbon content in such craters, and thus we can still tag the residues as organic in nature. However, we can neither identify their particular original nature, nor do something as simple as separating aromatic from aliphatic.

This result is disappointing. More work is required to test a wider range of organic materials to see how universal this result is. Work can also be done to change target types to see if the shock pressure can be lowered sufficiently to make a difference to the outcome of the capture process [44]. Although

it should be noted, that even at these speeds, whilst impact into porous media such as aerogel [45], lowers the peak shock pressures from 10–100 GPa on metals [29] down to hundreds of MPa [46], the capture process still results in processing of dust grains by ablation [43]. Finally, the work should be repeated as a function of impact speed to see if impacts at a lower speed can produce un-melted impact residue, which would in turn produce a definitive result separating the organic materials from each other. Once lower speed samples are obtained, which contain identifiable un-melted residue, it will be possible to explore the applicability of a wider range of analysis techniques. As it stands, the results suggest that a mission that flies past Enceladus collecting samples at speeds of 5 km s$^{-1}$ or above, will not produce un-melted macroscopic organic impact residues from impacts on solid aluminium metal targets, unless the organic has a very high melting point. This is of significance, as the modern understanding for the emergence of complexity from pre-biotic chemistry, is built on liquid water, energy (chemical dis-equilibria) and organic molecules. Obtaining an inventory of the organic molecule content in a region of astrobiological interest is therefore important.

**Author Contributions:** Conceptualization, M.B.; methodology, M.B., K.H.; formal analysis, K.H.; writing—original draft preparation, M.B.; writing—review and editing, M.B., K.H. All authors have read and agreed to the published version of the manuscript.

**Funding:** This research was funded by STFC, grant number ST/N000854/1.

**Acknowledgments:** We thank M.J. Cole for firing the light gas gun. We thank S. Armes (Univ. Sheffield) for providing the microparticles used in the study.

**Conflicts of Interest:** The authors declare no conflict of interest. The funders had no role in the design of the study; in the collection, analyses, or interpretation of data; in the writing of the manuscript, or in the decision to publish the results.

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
