# Peer review of "Organic Molecules: Is It Possible to Distinguish Aromatics from Aliphatics Collected by Space Missions in High-Speed Impacts?"

_sci, doi:10.3390/sci2030056_

Round 1

Reviewer 1 Report

A broad class of aromatic hydrocarbons made up of fused benzene rings have been suggested to be important in prebiotic chemistry. While exact pathways are debated, these molecules can be potentially relevant to the formation of broad range of molecular species. In contrast, aliphatic molecules contain carbon and hydrogen atom in straight chains representing a part of lipids and identified in a number of environments including asteroids and comets. The paper by M. Burchell and K. Harriss reports the results of laboratory experiments to investigate whether the current analytical methods can discriminate between aromatic and aliphatic molecules in impact experiments as the way not to look for signatures of life, but to learn about processes producing different carbon bonds in Enceladus and other extraterrestrial bodies in the Solar system.

The introduction discusses the Drake equation and it is not clear what is the relation between intelligence, B2FH model and aromatics and aliphatics. However, many details of the major objectives of the proposed investigation have not been described. For example, why these molecules should be formed in icy environments of Enceladus and how they will specify the pathways of prebiotic chemistry? 

I think that the idea to use high-speed impact of molecules emanating from this body with a spacecraft’s metal plate to discriminate between aromatic and aliphatic molecules is generally interesting. However, the paper lacks many technical details before it can be recommended for publication

1. The experiment’s setup and analysis lack details in the flux of PS (C8H8) molecules (as a proxy of aromatic molecules) versus PMMA, C5H9O2) (as a proxy for aliphatic molecules).

2. It is not clear why aluminum target has been used in the experiments?

3. Figure 3 shows the the impact craters, but it is unclear whether what scans of the crater were used for Raman spectra of crater residues?

4. It is not clear how peaks on oxygen can inform us about the type of molecules as the autoxidation of aromatic molecules as well as contamination as mentioned by the authors can introduce the spectral signature. Also, oxidation of an aluminum plate itself can affect the result of the experiment as water vapor get exposed during a mission to Europa or Enceladus.

5. How many experiments have been preformed?

Also, no quantitative analysis of shock pressure and its impact on chemical bonds of used molecules has been performed to predict possible outcomes of described experiments. Peaks in the spectra presented in Figure 3 are not properly discussed and it is not clear what they represent. Also, why not other supporting methods including gas chromatography (GC) and GC-MS (mass spectrometry) have been used in the analysis to resolve the structures of residues.

Thus, it is not clear whether the reported results are surprising and “disappointing”. This is also not clear how these experiments propose a methodology to discriminate between aromatics and aliphatics in future space missions.  Thus, I would recommend the authors to perform a more thorough analysis of possible outcomes before raising the question of discriminating between two types of molecules in laboratory experiment. Thus, this paper cannot be recommended for publication before all above-mentioned points are highlighted and paper is substantially revised.

Author Response

A broad class of aromatic hydrocarbons made up of fused benzene rings have been suggested to be important in prebiotic chemistry. While exact pathways are debated, these molecules can be potentially relevant to the formation of broad range of molecular species. In contrast, aliphatic molecules contain carbon and hydrogen atom in straight chains representing a part of lipids and identified in a number of environments including asteroids and comets. The paper by M. Burchell and K. Harriss reports the results of laboratory experiments to investigate whether the current analytical methods can discriminate between aromatic and aliphatic molecules in impact experiments as the way not to look for signatures of life, but to learn about processes producing different carbon bonds in Enceladus and other extraterrestrial bodies in the Solar system. The introduction discusses the Drake equation and it is not clear what is the relation between intelligence, B2FH model and aromatics and aliphatics. However, many details of the major objectives of the proposed investigation have not been described. For example, why these molecules should be formed in icy environments of Enceladus and how they will specify the pathways of prebiotic chemistry? I think that the idea to use high-speed impact of molecules emanating from this body with a spacecraft’s metal plate to discriminate between aromatic and aliphatic molecules is generally interesting. However, the paper lacks many technical details before it can be recommended for publication 1. The experiment’s setup and analysis lack details in the flux of PS (C8H8) molecules (as a proxy of aromatic molecules) versus PMMA, C5H9O2) (as a proxy for aliphatic molecules). 2. It is not clear why aluminum target has been used in the experiments? 3. Figure 3 shows the the impact craters, but it is unclear whether what scans of the crater were used for Raman spectra of crater residues? 4. It is not clear how peaks on oxygen can inform us about the type of molecules as the autoxidation of aromatic molecules as well as contamination as mentioned by the authors can introduce the spectral signature. Also, oxidation of an aluminum plate itself can affect the result of the experiment as water vapor get exposed during a mission to Europa or Enceladus. 5. How many experiments have been preformed? Also, no quantitative analysis of shock pressure and its impact on chemical bonds of used molecules has been performed to predict possible outcomes of described experiments. Peaks in the spectra presented in Figure 3 are not properly discussed and it is not clear what they represent. Also, why not other supporting methods including gas chromatography (GC) and GC-MS (mass spectrometry) have been used in the analysis to resolve the structures of residues. Thus, it is not clear whether the reported results are surprising and “disappointing”. This is also not clear how these experiments propose a methodology to discriminate between aromatics and aliphatics in future space missions. Thus, I would recommend the authors to perform a more thorough analysis of possible outcomes before raising the question of discriminating between two types of molecules in laboratory experiment. Thus, this paper cannot be recommended for publication before all above-mentioned points are highlighted and paper is substantially revised. We thank the referees for their work General comments: All the referees noted and objected to a somewhat imbalanced paper with an overly long introduction. This was due to some confusion. The paper was based upon an hour-long open lecture, and (as one referee seems to have spotted) was originally written up for publication as a book chapter. The introduction was thus a deliberately broad discussion of astrobiology. Publication plans changed and the paper was then to appear in a special issue of a journal, but unfortunately the original style was retained. Change: The broad introduction has been removed and replaced by a more focussed introduction on the specific question addressed in the paper. Specific comments of referee 1 1. The experiment’s setup and analysis lack details in the flux of PS (C8H8) molecules (as a proxy of aromatic molecules) versus PMMA, C5H9O2) (as a proxy for aliphatic molecules). Response: We have now explained that we looked at 7 PS craters and 10 PMMA craters in this study 2. It is not clear why aluminum target has been used in the experiments? Response This has been widely used for impact residue studies and was the sub-strate used in the NASA Stardust mission to collect cometary dust grains. This is now stated in the text. 3. Figure 3 shows the the impact craters, but it is unclear whether what scans of the crater were used for Raman spectra of crater residues? Response; We have added text to make it clear we took spot spectra from various craters where the SEM work showed there was residue. 4. It is not clear how peaks on oxygen can inform us about the type of molecules as the autoxidation of aromatic molecules as well as contamination as mentioned by the authors can introduce the spectral signature. Also, oxidation of an aluminum plate itself can affect the result of the experiment as water vapor get exposed during a mission to Europa or Enceladus. Response: Agreed. That is why we pointed out that although O is expected in PMMA and not PS, the appearance of O as a contaminant means it is hard to build up an elemental composition of the impactor. Thus the ideal solution would be an analysis technique based on structure. 5. How many experiments have been preformed? Response; We have now explicity said now many craters were examined in each shot. Also, no quantitative analysis of shock pressure and its impact on chemical bonds of used molecules has been performed to predict possible outcomes of described experiments. Response: We have added an estimate of peak shock pressure. We have also added a calculation of temperature increase in the samples, and compared it to the melting point of the samples. Peaks in the spectra presented in Figure 3 are not properly discussed and it is not clear what they represent. Response: We have improved the notation on the spectra. Also, why not other supporting methods including gas chromatography (GC) and GC-MS (mass spectrometry) have been used in the analysis to resolve the structures of residues. Response: Given that we have found melt and observed that the original structure has been lost we did not go further with the current samples. However, when we repeat the work at lower speeds, and hopefully find a speed where some material is not melted, we will then explore how to remove reside from the craters and present it for a wider chain of analysis. This comment has been added to the ms.

Reviewer 2 Report

Dear authors,

Your manuscript is interesting and easy to read. Nevertheless, I feel that your introduction is too broad, sometimes far from your research question.

In my opinion, you should re-write this manuscript in order to submit it as a short paper, in a more assertive way.

Your introduction should focus more on what you write in the first couple of paragraphs of your discussion. And your discussion should clarify the relevance of your assays in a more direct way and refer to the results obtained.

Your introduction would read very well as a book chapter on a more general topic.

I hope my comments are useful.

Author Response

Dear authors, Your manuscript is interesting and easy to read. Nevertheless, I feel that your introduction is too broad, sometimes far from your research question. In my opinion, you should re-write this manuscript in order to submit it as a short paper, in a more assertive way. Your introduction should focus more on what you write in the first couple of paragraphs of your discussion. And your discussion should clarify the relevance of your assays in a more direct way and refer to the results obtained. Your introduction would read very well as a book chapter on a more general topic. I hope my comments are useful. We thank the referees for their work General comments: All the referees noted and objected to a somewhat imbalanced paper with an overly long introduction. This was due to some confusion. The paper was based upon an hour-long open lecture, and (as one referee seems to have spotted) was originally written up for publication as a book chapter. The introduction was thus a deliberately broad discussion of astrobiology. Publication plans changed and the paper was then to appear in a special issue of a journal, but unfortunately the original style was retained. Change: The broad introduction has been removed and replaced by a more focussed introduction on the specific question addressed in the paper.

Reviewer 3 Report

Dear authors,

I have read with great attention your manuscript entitled “Organic Molecules: Is It Possible to Distinguish Aromatics from Aliphatics Collected by Space Missions in High Speed Impacts?” submitted for publication in Sci.

Unfortunately, I think this article is not suitable for publication for several reasons, the most important among which are:

- The relevance of the samples used – 20 µm diameter spheres of polystyrene and poly(methyl methacrylate) – is very poor. The fact that these samples are destroyed during the impact does not demonstrate anything regarding the potential degradation of other organic molecules entrapped within Enceladus icy particles. The physical properties and the nature the samples are crucial. The sample mass is also of primary importance, not only the speed, since the kinetic energy is a key parameter. 

- The way of writing is not corresponding to a scientific publication; it is rather a spoken style.

- The article is badly structured: the introduction is 4 pages (approx. 200 lines), the materials and methods are less than 20 lines, results are only 15 lines and the discussion is only 1 page (approx. 50 lines).

- The first two pages (first part of the introduction) are totally out of the scope of the article, with no link to the experiments.

This study requires further investigations, with complementary experiments and analyses, and the manuscript must be fully reorganised prior resubmission, therefore I do not provide a detailed review including line by line comments.

Best regards

Author Response

Dear authors, I have read with great attention your manuscript entitled “Organic Molecules: Is It Possible to Distinguish Aromatics from Aliphatics Collected by Space Missions in High Speed Impacts?” submitted for publication in Sci. Unfortunately, I think this article is not suitable for publication for several reasons, the most important among which are: - The relevance of the samples used – 20 µm diameter spheres of polystyrene and poly(methyl methacrylate) – is very poor. The fact that these samples are destroyed during the impact does not demonstrate anything regarding the potential degradation of other organic molecules entrapped within Enceladus icy particles. The physical properties and the nature the samples are crucial. The sample mass is also of primary importance, not only the speed, since the kinetic energy is a key parameter. - The way of writing is not corresponding to a scientific publication; it is rather a spoken style. - The article is badly structured: the introduction is 4 pages (approx. 200 lines), the materials and methods are less than 20 lines, results are only 15 lines and the discussion is only 1 page (approx. 50 lines). - The first two pages (first part of the introduction) are totally out of the scope of the article, with no link to the experiments. This study requires further investigations, with complementary experiments and analyses, and the manuscript must be fully reorganised prior resubmission, therefore I do not provide a detailed review including line by line comments. Best regards We thank the referees for their work General comments: All the referees noted and objected to a somewhat imbalanced paper with an overly long introduction. This was due to some confusion. The paper was based upon an hour-long open lecture, and (as one referee seems to have spotted) was originally written up for publication as a book chapter. The introduction was thus a deliberately broad discussion of astrobiology. Publication plans changed and the paper was then to appear in a special issue of a journal, but unfortunately the original style was retained. Change: The broad introduction has been removed and replaced by a more focussed introduction on the specific question addressed in the paper. Regarding the comment that energy is the key, this is not quite right. The energy density is what is important. Further, the mass is constrained, as the particles which may be present in the plumes are limited in size, so we can't just imagine large particles.

Reviewer 4 Report

Very minor grammar edit suggested as highlighted on MS

Author Response

Referee 4 made suggestions re typos and minor English usage on a pdf copy of the ms. Referee 4 made suggestions on a pdf copy of the ms. We have incorporated these. A summary follows: Line 41 – comma deleted Line 43 – comma deleted (also see Referee 3) Line 51 – should “characteristic” be plural? We don’t think so. Line 64 – comma replaced by colon (also see Referee 3) The use of “in situ” appeara again, saying it should be in italics, but on our version it already is. Line 104 – comma deleted Line 133 Repeated words deleted Line 256 – comma added

Round 2

Reviewer 1 Report

The revised paper reflects the referee's comments and suggestions. I would recommend to add that apart with the "the mantra-find the water" mentioned in the introduction, authors should add that energy source is also required to promote prebiotic chemistry that further complexity of biologically relevant molecules as evidenced from experiments and theory.

Author Response

The revised paper reflects the referee's comments and suggestions. I would recommend to add that apart with the "the mantra-find the water" mentioned in the introduction, authors should add that energy source is also required to promote prebiotic chemistry that further complexity of biologically relevant molecules as evidenced from experiments and theory. We have added text at the in the introduction and conclusion pointing out that the search for life involves more than just water and includes chemical dis-equilibria and organic molecules. Lines 48 – 52 and 311 – 315

Reviewer 3 Report

Dear authors

Thank you for submitting a revised version of your manuscript entitled « Organic Molecules: Is It Possible To Distinguish Aromatics From Aliphatics Collected By Space Missions in High-Speed Impacts?” for publication in the journal Sci.

Your responses to the reviewers answer most of the criticisms formulated. Moreover, you entirely rewrote the manuscript to take into account comments and recommendations. This new version is clear, the introduction is consistent with the experiments and the objectives are well defined.

Nevertheless, I still have some comments and suggest minor revisions prior publication:

  • Page 1, Abstract l. 6: “5 km s−1 and above” is a little bit vague. It would be better to give a range.
  • Page 1, Introduction, l. 5: “identify” instead of “identity”
  • Page 2: replace “in-situ” and “in situ” by “in situ”, in italic (4 occurrences)
  • Page 3, in the middle: “For example, Raman spectrometers have been made robust enough to be deployed on space missions and one will soon be sent to the surface of Mars (ESA’s ExoMars rover mission).” In fact, two Raman spectrometers will be on-board the NASA’s Mars 2020 rover and one on-board the ESA’s ExoMars 2022 rover.
  • Page 5, Results, l.3: a quotation mark is after “”breathing mode”
  • Page 5, Results, paragraph 2, l.2: “For both PS and PMMA, Raman spectra have been heavily altered from that of the raw materials, with the Raman D and G bands for carbon dominating.” It would be good here to add some information regarding the D and G bands, something like “These two bands, respectively referring to “disorder” and “graphite” bands, are characteristic of disordered carbonaceous material such as kerogen, soot or burned paper (Pasteris and Wopenka, Astrobiology, 3:4, 727-738, 2003). This Raman signal is typical of carbonized organic matter. ”
  • Page 6, Discussion, same paragraph: remove the equations from the text and display them as:

y = ax + b                                            (1)

  • page 6, Discussion, first paragraph: It would be good to develop the sentences “We estimate the peak pressure using the Planar Impact Approximation [39].This uses a linear wave speed equation for all the materials involved, of the form U = C + Su, where U is shock speed and u is particle speed.” and write something like “Based on the Planar Impact Approximation [39], we estimate the peak pressure using the following equation:

      XXXXXX                                              (1)

              where

                                                U = C + Su                                         (2)

is the linear wave speed equation for all the materials involved, with U the shock speed and u the particle speed.”

  • Page 6, Discussion, same paragraph: the values of C and S are missing.
  • Page 6, Discussion, second paragraph, l.1: replace “that is” by “;”.
  • Page 6, Discussion, same paragraph: I have to confess that I am not a specialist in physics of impacts; however, this part seems to be relatively simple. In particular, in the second paragraph, you estimate the temperature increase using the equation of isobaric process normally applying to reversible transformations. Could you add a reference in which this equation is used for impact processes? I mentioned in my first review that the sample mass is of primary importance, not only the speed, since the kinetic energy is a key parameter. I agree with you that it is the energy density that is important. I wanted to say that, for a given volume, the mass is important. Indeed, I continue to think that the density of the materials is crucial in the impact phenomenon. For a given volume of the impactor, the kinetic energy increases with the density. Similarly, for a given mass, the size of the impactor increases with the deceasing density and the pressure during impact decreases. The mechanical and physical properties of the target should also play a role. I understand that your calculation of the temperature increase based only on the speed and on the specific heat capacity of the sphere is an approximation but some details regarding the limits of the model used are required.
  • Page 7, l.1 : “This suggests that the simple models either over-estimate the peak post-shock temperature, or the whole impacting grain is not uniformly shocked, leaving macroscopic regions un-melted.” I agree with you but I also think it is because your model does not take into account the latent heat involved in phase changes.
  • Page 7, last paragraph: “Finally, the work should be repeated as a function of impact speed to see if impacts at a lower speed can produce un-melted impact residue, which would in turn produce a definitive result separating the organic materials from each other.” If it is possible, I think it would be interesting to make tests with different impactor morphologies.

Best regards

Frédéric Foucher

Author Response

Page 1, Abstract l. 6: “5 km s−1 and above” is a little bit vague. It would be better to give a range. Page 1, Introduction, l. 5: “identify” instead of “identity” Page 2: replace “in-situ” and “in situ” by “in situ”, in italic (4 occurrences) Page 3, in the middle: “For example, Raman spectrometers have been made robust enough to be deployed on space missions and one will soon be sent to the surface of Mars (ESA’s ExoMars rover mission).” In fact, two Raman spectrometers will be on-board the NASA’s Mars 2020 rover and one on-board the ESA’s ExoMars 2022 rover. Page 5, Results, l.3: a quotation mark is after “”breathing mode” Page 5, Results, paragraph 2, l.2: “For both PS and PMMA, Raman spectra have been heavily altered from that of the raw materials, with the Raman D and G bands for carbon dominating.” It would be good here to add some information regarding the D and G bands, something like “These two bands, respectively referring to “disorder” and “graphite” bands, are characteristic of disordered carbonaceous material such as kerogen, soot or burned paper (Pasteris and Wopenka, Astrobiology, 3:4, 727-738, 2003). This Raman signal is typical of carbonized organic matter. ” Page 6, Discussion, same paragraph: remove the equations from the text and display them as: y = ax + b (1) page 6, Discussion, first paragraph: It would be good to develop the sentences “We estimate the peak pressure using the Planar Impact Approximation [39].This uses a linear wave speed equation for all the materials involved, of the form U = C + Su, where U is shock speed and u is particle speed.” and write something like “Based on the Planar Impact Approximation [39], we estimate the peak pressure using the following equation: XXXXXX (1) where U = C + Su (2) is the linear wave speed equation for all the materials involved, with U the shock speed and u the particle speed.” Page 6, Discussion, same paragraph: the values of C and S are missing. Page 6, Discussion, second paragraph, l.1: replace “that is” by “;”. Page 6, Discussion, same paragraph: I have to confess that I am not a specialist in physics of impacts; however, this part seems to be relatively simple. In particular, in the second paragraph, you estimate the temperature increase using the equation of isobaric process normally applying to reversible transformations. Could you add a reference in which this equation is used for impact processes? I mentioned in my first review that the sample mass is of primary importance, not only the speed, since the kinetic energy is a key parameter. I agree with you that it is the energy density that is important. I wanted to say that, for a given volume, the mass is important. Indeed, I continue to think that the density of the materials is crucial in the impact phenomenon. For a given volume of the impactor, the kinetic energy increases with the density. Similarly, for a given mass, the size of the impactor increases with the deceasing density and the pressure during impact decreases. The mechanical and physical properties of the target should also play a role. I understand that your calculation of the temperature increase based only on the speed and on the specific heat capacity of the sphere is an approximation but some details regarding the limits of the model used are required. Page 7, l.1 : “This suggests that the simple models either over-estimate the peak post-shock temperature, or the whole impacting grain is not uniformly shocked, leaving macroscopic regions un-melted.” I agree with you but I also think it is because your model does not take into account the latent heat involved in phase changes. Page 7, last paragraph: “Finally, the work should be repeated as a function of impact speed to see if impacts at a lower speed can produce un-melted impact residue, which would in turn produce a definitive result separating the organic materials from each other.” If it is possible, I think it would be interesting to make tests with different impactor morphologies. • Page 1, Abstract l. 6: “5 km s−1 and above” is a little bit vague. It would be better to give a range. • Response: The main CDA paper, Postberg et al, Nature 2018, contains impact ionisation mass spectra from the encounter with the Enceladus plume in the speed range 5 – 15 km/s, so this range is now used in the abstract. See line 20 • Page 1, Introduction, “identify” instead of “identity” • Response: Corrected on line 45 • Page 2: replace “in-situ” and “in situ” by “in situ”, in italic (4 occurrences) • Response: Done, see lines 64, 90, 100 and 102 • Page 3, in the middle: “For example, Raman spectrometers have been made robust enough to be deployed on space missions and one will soon be sent to the surface of Mars (ESA’s ExoMars rover mission).” In fact, two Raman spectrometers will be on-board the NASA’s Mars 2020 rover and one on-board the ESA’s ExoMars 2022 rover. • Response: This has been updated. See line 130 • Page 5, Results, l.3: a quotation mark is after “”breathing mode” • Response: This was indeed missing and has been added, see line 232 • Page 5, Results, paragraph 2, l.2: “For both PS and PMMA, Raman spectra have been heavily altered from that of the raw materials, with the Raman D and G bands for carbon dominating.” It would be good here to add some information regarding the D and G bands, something like “These two bands, respectively referring to “disorder” and “graphite” bands, are characteristic of disordered carbonaceous material such as kerogen, soot or burned paper (Pasteris and Wopenka, Astrobiology, 3:4, 727-738, 2003). This Raman signal is typical of carbonized organic matter. ” • Response: We have added the suggested text. See lines 241 – 244. Also see line 429 for the new reference, ref [39]. Subsequent references have had to be renumbered in the ms. • Page 6, Discussion, same paragraph: remove the equations from the text and display them as: y = ax + b (1) Response Done, see next comment • page 6, Discussion, first paragraph: It would be good to develop the sentences “We estimate the peak pressure using the Planar Impact Approximation [39].This uses a linear wave speed equation for all the materials involved, of the form U = C + Su, where U is shock speed and u is particle speed.” and write something like “Based on the Planar Impact Approximation [39], we estimate the peak pressure using the following equation: XXXXXX (1) where U = C + Su (2) is the linear wave speed equation for all the materials involved, with U the shock speed and u the particle speed.” Response: We have adjusted the text along the lines suggested. See lines 257 - 262 • Page 6, Discussion, same paragraph: the values of C and S are missing. • Response: Sorry, we had missed this in the proof reading. The values are now added and the peak shock pressure for PMMA has been corrected (we noticed we had originally given it for a lower speed impact). See lines 263 - 265 • Page 6, Discussion, second paragraph, l.1: replace “that is” by “;” • Response: Changed, see line 267 • Page 6, Discussion, same paragraph: I have to confess that I am not a specialist in physics of impacts; however, this part seems to be relatively simple. In particular, in the second paragraph, you estimate the temperature increase using the equation of isobaric process normally applying to reversible transformations. Could you add a reference in which this equation is used for impact processes? I mentioned in my first review that the sample mass is of primary importance, not only the speed, since the kinetic energy is a key parameter. I agree with you that it is the energy density that is important. I wanted to say that, for a given volume, the mass is important. Indeed, I continue to think that the density of the materials is crucial in the impact phenomenon. For a given volume of the impactor, the kinetic energy increases with the density. Similarly, for a given mass, the size of the impactor increases with the deceasing density and the pressure during impact decreases. The mechanical and physical properties of the target should also play a role. I understand that your calculation of the temperature increase based only on the speed and on the specific heat capacity of the sphere is an approximation but some details regarding the limits of the model used are required. • Response: This is indeed a simple approximation, but is used in the literature, a reference has been added. Peak pressure depends on several parameters, and the Planar Impact approximation for example includes the density of the materials so it is indeed important. Some extra text has been added pointing out some of the limitations. See lines 272 - 277 • Page 7, l.1 : “This suggests that the simple models either over-estimate the peak post-shock temperature, or the whole impacting grain is not uniformly shocked, leaving macroscopic regions un-melted.” I agree with you but I also think it is because your model does not take into account the latent heat involved in phase changes. • Response: We have made the caveat clearer. See line 288 • Page 7, last paragraph: “Finally, the work should be repeated as a function of impact speed to see if impacts at a lower speed can produce un-melted impact residue, which would in turn produce a definitive result separating the organic materials from each other.” If it is possible, I think it would be interesting to make tests with different impactor morphologies. • Response: There are lots of things we would like to test in future work. As we say in the paper, different target types, different projectile types, different impact speeds…are all of interest and we have started looking at them. Hence we have added a reference to a paper submitted since this one was originally written, and which indeed starts to vary target type. See line 301, and new ref [44] lines 438 - 441. All comments are in the first box All responses are in the first reply

Round 3

Reviewer 3 Report

Dear authors

Thank you for having considered my comments and suggestions. I think your manuscript can now be accepted for publication. I have just noticed some very small typo errors.

  • Page 1, Abstract: there are useless brackets in “poly(methylmethacrylate)”
  • Page 1, Introduction, l. 3: useless coma after “for life”
  • Page 2, last paragraph before section “Enceladus and Sampling its Plumes”: better to write “One simple question is:” rather than “One simple question, is”
  • “in situ” normally writes in italic
  • Page 3, in the middle: “For example, Raman spectrometers have been made robust enough to be deployed on space missions and one will soon be sent to the surface of Mars (on NASA’s Mars 2020 mission and ESA’s future ExoMars rover mission).” Replace “one” by “two”
  • Page 4, l.4: a laser at 534 nm is not common, it is generally 532 nm. Is it an error?

Best regards

Frédéric Foucher

Author Response

Page 1, Abstract: there are useless brackets in “poly(methylmethacrylate)” Page 1, Introduction, l. 3: useless coma after “for life” Page 2, last paragraph before section “Enceladus and Sampling its Plumes”: better to write “One simple question is:” rather than “One simple question, is” “in situ” normally writes in italic Page 3, in the middle: “For example, Raman spectrometers have been made robust enough to be deployed on space missions and one will soon be sent to the surface of Mars (on NASA’s Mars 2020 mission and ESA’s future ExoMars rover mission).” Replace “one” by “two” Page 4, l.4: a laser at 534 nm is not common, it is generally 532 nm. Is it an error? We thank the referee for the careful reading and make changes as follows • Page 1, Abstract: there are useless brackets in “poly(methylmethacrylate)” Action: Line 31, change made • Page 1, Introduction, l. 3: useless coma after “for life” • Action: Line 43, comma deleted • Page 2, last paragraph before section “Enceladus and Sampling its Plumes”: better to write “One simple question is:” rather than “One simple question, is” • Action: Line 64, change made • “in situ” normally writes in italic • Action: We don’t understand, it is in italics. I have done a search and can find no in normal font. • Page 3, in the middle: “For example, Raman spectrometers have been made robust enough to be deployed on space missions and one will soon be sent to the surface of Mars (on NASA’s Mars 2020 mission and ESA’s future ExoMars rover mission).” Replace “one” by “two” • Action: Change made, line 129 • Page 4, l.4: a laser at 534 nm is not common, it is generally 532 nm. Is it an error? • Action: We keep fixing this typo and it keeps creeping back in, change made. Line 197